# Observation of magnetic adatom-induced Majorana vortex and its hybridization with field-induced Majorana vortex in an iron-based superconductor

Peng Fan[1,2,7], Fazhi Yang[1,2,7], Guojian Qian[1,2,7], Hui Chen [1,2,3,7], Yu-Yang Zhang [1,2,4], Geng Li [1,2,4], Zihao Huang[1,2], Yuqing Xing [1,2], Lingyuan Kong [1,2], Wenyao Liu [1,2], Kun Jiang[1,5], Chengmin Shen[1,2,4], Shixuan Du[1,2,4], John Schneeloch[6], Ruidan Zhong[6], Genda Gu[6], Ziqiang Wang[5✉], Hong Ding [1,3,4✉] & Hong-Jun Gao [1,2,3,4✉]

Braiding Majorana zero modes is essential for fault-tolerant topological quantum computing. Iron-based superconductors with nontrivial band topology have recently emerged as a surprisingly promising platform for creating distinct Majorana zero modes in magnetic vortices in a single material and at relatively high temperatures. The magnetic field-induced Abrikosov vortex lattice makes it difficult to braid a set of Majorana zero modes or to study the coupling of a Majorana doublet due to overlapping wave functions. Here we report the observation of the proposed quantum anomalous vortex with integer quantized vortex core states and the Majorana zero mode induced by magnetic Fe adatoms deposited on the surface. We observe its hybridization with a nearby field-induced Majorana vortex in iron-based superconductor $FeTe_{0.55}Se_{0.45}$. We also observe vortex-free Yu-Shiba-Rusinov bound states at the Fe adatoms with a weaker coupling to the substrate, and discover a reversible transition between Yu-Shiba-Rusinov states and Majorana zero mode by manipulating the exchange coupling strength. The dual origin of the Majorana zero modes, from magnetic adatoms and external magnetic field, provides a new single-material platform for studying their interactions and braiding in superconductors bearing topological band structures.

[1] Beijing National Laboratory for Condensed Matter Physics and Institute of Physics, Chinese Academy of Sciences, Beijing, China. [2] School of Physical Sciences, University of Chinese Academy of Sciences, Beijing, China. [3] Songshan Lake Materials Laboratory, Dongguan, Guangdong, China. [4] CAS Center for Excellence in Topological Quantum Computation, University of Chinese Academy of Sciences, Beijing, China. [5] Department of Physics, Boston College, Boston, MA, USA. [6] Condensed Matter Physics and Materials Science Department, Brookhaven National Laboratory, Upton, NY, USA. [7] These authors contributed equally: Peng Fan, Fazhi Yang, Guojian Qian, Hui Chen. ✉email: wangzi@bc.edu; dingh@iphy.ac.cn; hjgao@iphy.ac.cn

The band structure of iron-based superconductor FeTe$_{0.55}$Se$_{0.45}$ has a nontrivial Z$_2$ topological invariant and supports helical Dirac fermion topological surface states (TSS)[1,2], which was confirmed recently by spin-resolved and angle-resolved photoemission spectroscopy[3]. Remarkably, below the bulk transition temperature $T_c$, superconducting (SC) TSS were observed with a pairing gap[3]. This makes FeTe$_{0.55}$Se$_{0.45}$ a novel single-material platform for generating Majorana zero modes (MZMs) at the ends of a vortex line[2,4,5]. Recently, strong evidence of MZMs inside the magnetic field-induced vortices have been observed by scanning tunneling microscope/spectroscopy (STM/S) on the vortex lattice of this type-II superconductor[6–9].

This Majorana platform also presents new challenges for the basic understanding of defect excitations in superconductors with a topological nontrivial band structure, and new possibilities for creating MZMs under different physical conditions. In general, superconductors can host two kinds of defect excitations as in-gap bound states: the Yu-Shiba-Rusinov (YSR) states[10–17] localized at a magnetic impurity and the Caroli-de-Gennes-Matricon (CdGM) states[18–26] inside a magnetic vortex core. To date, these excitations have appeared distinctly in nature; a magnetic impurity induces the YSR states carrying spin, whereas an external magnetic field creates vortices of the whirling supercurrents. In stark contrast to the traditional YSR states, robust zero-bias peaks (ZBPs), sharing the quintessential spectroscopic properties of MZMs, were observed by STM/S at the magnetic interstitial Fe impurities in FeTe$_{0.55}$Se$_{0.45}$[27], but without applying a magnetic field. A recent theoretical proposal[28] attributes the observed ZBP to a MZM bound to a quantum anomalous vortex (QAV) nucleated spontaneously at the magnetic Fe atom. The role of the magnetic field is played by the exchange coupling of the spin and orbital moment of the Fe impurity located at the C$_4$ symmetric sites, which generates circulating supercurrents by the spin–orbit coupling and modulates the phase of the superconducting order parameter. When the exchange coupling is strong enough, a condition favored by the very small Fermi energy (~5 meV) in FeTe$_{0.55}$Se$_{0.45}$[3], a transition from the vortex-free YSR states to the QAV was predicted to take place[28]. A MZM emerges inside the QAV core from the superconducting TSS, since the Berry phase of the Dirac fermions transforms the total angular momentum quantum number of the CdGM vortex core states[18] into integers, naturally supporting a zero-energy bound state[8,28].

## Results

### Characterizations of samples and Fe adatoms.

To probe the remarkable nature of defect excitations in the superconducting TSS, we deposit magnetic Fe adatoms on the cleaved (001) surface of a single crystal of FeTe$_{0.55}$Se$_{0.45}$ (Fig. 1a, b) with the substrate temperature below 20 K. Before depositing the Fe adatoms, we scan the surface to ensure that there is no interstitial Fe adatom (see Supplementary Note 1), to avoid the mix of interstitial Fe adatoms and deposited Fe adatoms. In contrast to the growth-induced interstitial Fe impurity in the bulk, the adsorbed Fe adatoms are distributed at various heights above the surface and in different planar locations with respect to the C$_4$ symmetry sites. As a result, the magnetic Fe adatoms with varying exchange couplings reveal much richer phenomena of the defect excitations of the superconducting TSS.

The STM image of a surface region after the atomic deposition (Fig. 1c) shows scattered Fe adatoms as the bright spots with a coverage of ~0.04%. The zero-energy d$I$/d$V$ map in the same area of Fig. 1c, displays relatively high density of states at the locations of the Fe adatoms (Fig. 1d), consistent with the physical picture that magnetic Fe impurities generate in-gap states. From the statistics of >100 measurements (Supplementary Note 1 and Supplementary Fig. 2), we identify two types of in-gap states

localized around the Fe adatoms with distinct d$I$/d$V$ spectra exemplified in Fig. 1e. The type-I adatoms, which represent about 10% of our measurements, exhibit a sharp ZBP reminiscent of a MZM coexisting with other in-gap states in the d$I$/d$V$ spectrum. In contrast, the conductance spectrum of the type-II adatoms, which represent about 90% of our measurements, shows the YSR states, featuring a pair of in-gap states at particle-hole symmetric peak energy positions, but with asymmetric peak weights. In comparison, the typical d$I$/d$V$ spectrum on the clean surface without the adatom shows a hard superconducting gap without in-gap states (Fig. 1e).

### QAV states in type-I Fe adatoms.

We begin with the type-I Fe adatoms. A high-resolution topographic image (Fig. 2a) shows an isolated type-I adatom. A circular pattern appears in the zero-energy d$I$/d$V$ map in the vicinity of the Fe site (Fig. 2b), with the zero-energy intensity center slightly offset from the Fe site. The breaking of the rotation symmetry is likely due to the canting of the magnetic moment of the Fe adatom away from the surface normal and the presence of spin–orbit coupling. The waterfall-like d$I$/d$V$ spectra (Fig. 2c) and the intensity plot (Fig. 2d) along the red dashed line-cut in Fig. 2a clearly resolve the ZBP and other peaks at nonzero energies inside the superconducting gap. The ZBP persists to about 2.5 nm from the center, indicating the existence of a localized zero-energy state bound to the Fe adatom with the spatial extent comparable to that of the MZM in a magnetic field-induced vortex core in FeTe$_{0.55}$Se$_{0.45}$[6]. To explore the nature of the discretized in-gap states, we extract several d$I$/d$V$ spectra from Fig. 2c and show them as a stacking plot in Fig. 2e. Doing so accounts for the spatial distributions of the in-gap states and the sample inhomogeneity[6,7] due to Te/Se alloying, which has proven to be useful for studying the core states of magnetic field-induced vortices in FeTe$_{0.55}$Se$_{0.45}$[8]. The sequence of discretized bound states, including the zero-energy state, is clearly visible as the pronounced peaks labeled by L$_0$, L$_{\pm1}$, L$_{\pm2}$. The energy positions of the conductance peaks (L$_n$) are plotted in Fig. 2f. Intriguingly, the average energies (solid lines) of the discrete quantum states bound to the Fe adatom follow closely a sequence of integer quantization $E_n \approx n\varepsilon$, $n = 0, \pm1, \pm2, \ldots$, with the minigap $\varepsilon \sim$ 1.0 meV, the same integer sequence (with a minigap ~0.6 meV) followed by the quantized vortex core states observed recently[8] in magnetic field-induced vortices that host the MZM[6,7]. The different values of the minigap are caused by the fluctuation of the superconducting gap $\Delta$ and Fermi energy $E_f$ on different surface, due to the inhomogeneous Te and Se atoms in FeTe$_{0.55}$Se$_{0.45}$. More cases are shown in Supplementary Fig. 3 for the type-I Fe adatoms. Such an integer quantized sequence is the hallmark of the CdGM vortex core states of the superconducting TSS in the quantum limit[8,28]. Our observations thus provide substantiated and compelling evidence that vortex-like topological defect excitations such as the QAVs nucleate spontaneously at the type-I Fe adatoms and the ZBP corresponds to a vortex MZM.

It is necessary to check the temperature and magnetic field dependence of the ZBP, since a vortex MZM would respond differently than vortex-free defect states. The temperature evolution of the ZBP on the Fe adatom turns out to be very similar to that of the MZM in a field-induced vortex[6]. The ZBP intensity of the center spectrum pointed by the black arrow in Fig. 2c decreases with increasing temperatures, and becomes almost invisible at 4.2 K (Fig. 2g). The magnetic field dependence is also very similar to that of a vortex MZM as the ZBP does not split or broaden for fields up to 8 T (Fig. 2h), provided that no field-induced vortices enter the region near the adatom when the magnetic field is applied. The ZBP is, however, sensitive to the location of the adsorbed Fe adatom. We found that all type-I

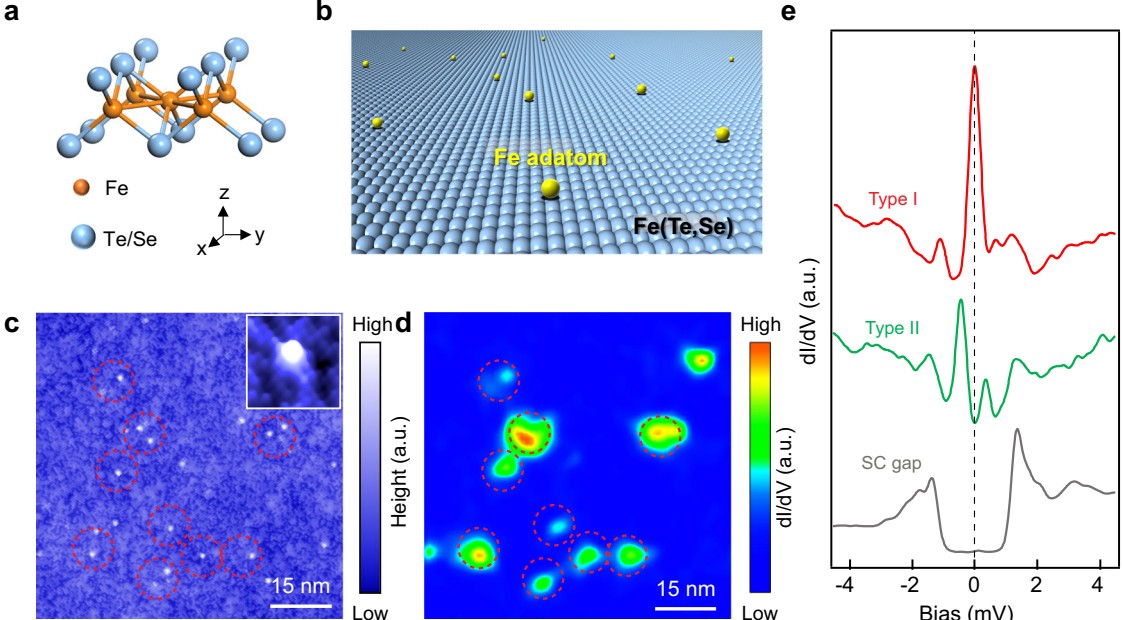

**Fig. 1 Characterization of deposited Fe adatoms on FeTe$_{0.55}$Se$_{0.45}$ surface. a** Crystal structure of Fe(Te,Se). **b** Schematics of Fe adatoms deposited on the surface of FeTe$_{0.55}$Se$_{0.45}$. The adsorption sites are random, including both on and off of the C$_4$ symmetric sites in the center of four Te/Se atoms and at various heights above the surface (Supplementary Note 1). **c** A STM image ($V_s = -10$ mV, $I_t = 100$ pA) after atomic Fe atom deposition. The bright dots correspond to Fe adatoms with a coverage of 0.04%. Inset: A 4 × 4 nm atomic-resolution topographic image showing a single-Fe adatom located at a high-symmetry site. **d** A d$I$/d$V$ map ($V_s = -10$ mV, $I_t = 100$ pA) of **c** at zero energy. The high conductance areas and the Fe adatom positions are in good spatial correspondence, as highlighted by the red dashed circles. **e** Typical d$I$/d$V$ spectra ($V_s = -10$ mV, $I_t = 200$ pA) showing the zero-bias peak on type-I Fe adatoms (red curve), YSR states (green curve) on type-II Fe adatoms, and the clean superconducting (SC) gap away from the adatoms (gray curve).

adatoms producing integer quantized bound states anchored by the ZBP are adsorbed at the high-symmetry sites in the center of four Te/Se atoms. To test the robustness of this finding, we manipulate a type-I adatom by the STM tip to a different location away from the C$_4$ symmetric site. The ZBP disappears and a pair of in-gap state at nonzero energy emerges (Supplementary Note 3 and Supplementary Fig. 4). After annealing the sample to 15 K and performing the measurement again at 0.4 K, the adatom diffuses back to its original high-symmetry site and the ZBP reappears. Thus, the high-symmetry site is a prerequisite for the induced ZBP, which agrees with the proposed theory that the orbital magnetic moment of the Fe adatoms at C$_4$ symmetric locations plays an important role for the nucleation of the QAV[28].

**YSR states in type-II Fe adatoms.** Next, we turn to study the type-II Fe adatoms (Supplementary Note 4). In contrast to the type-I adatoms, the conductance exhibits predominantly a pair of in-gap peaks at nonzero energies without the ZBP. Applying an external magnetic field, we observe that the peaks shift approximately linearly to higher energies (Supplementary Fig. 5d) away from the Fermi level, consistent with a pair of spin-polarized YSR in-gap states. We find that the type-II Fe adatoms are adsorbed at myriad locations on or off the high-symmetry axis and induce YSR states at different energies, indicative of broadly varying exchange couplings to the superconducting quasiparticles. In special cases, we also observe YSR states located very close to zero energy, which nevertheless split under the magnetic field (Supplementary Note 4 and Supplementary Fig. 6) and are, therefore, distinct from the robust ZBP observed on the type-I Fe adatoms.

**Reversible transition between YSR states and MZMs in some type-II Fe adatoms.** These observations motivate us to manipulate

the exchange coupling between the magnetic adatoms and the substrate by tuning the tip to sample distance[29,30]. In the approaching-tip process, the STM tip needs to be positioned on top of the Fe adatom, which makes it impossible to acquire spatial maps while keeping the position of the Fe adatom frozen. The electrostatic force of an approaching-tip can prod and move the Fe adatom in directions parallel and perpendicular to the surface (Fig. 3a), which can affect the atomic orbital moment of the Fe adatom and the spin–orbit exchange coupling to the superconductor. In STM/S, the tunnel-barrier conductance $G_N \equiv I_t/V_s$, where $I_t$ is the tunneling current and $V_s$ is the bias voltage, governs the tunnel coupling and changes with the tip-sample distance. Performing tunneling conductance measurements as a function of $G_N$, we find that the energies of the YSR states are modulated ubiquitously when the tip approaches the type-II Fe adatoms (Supplementary Note 5). The observed crossing and reversal of the in-gap states (Supplementary Fig. 7), a trademark of the YSR states, confirms that reducing the tip to sample distance monotonically increases the exchange coupling between the Fe atom and the superconductor.

Unexpectedly, as the STM tip approaches a significant number of the type-II Fe adatoms (~27%), the pair of YSR states modulates with increasing $G_N$, but then coalesces into a single ZBP in the waterfall plot of d$I$/d$V$ spectra (Fig. 3c) and the intensity plot (Fig. 3d), which remains robust under further increase of the barrier conductance. Note that the emergence of the ZBP out of the YSR states is different from the transition point between the screened spin-singlet and doublet-ground states[29,31], where the two YSR states are approximately degenerate at zero energy as marked by the red arrow in Fig. 3d. To probe the change in the nature of the in-gap states with different barrier conductance, we repeated the entire process under an applied magnetic field. The d$I$/d$V$ spectra and the intensity plot obtained under 6 T (Fig. 3e, f) show that the vortex-

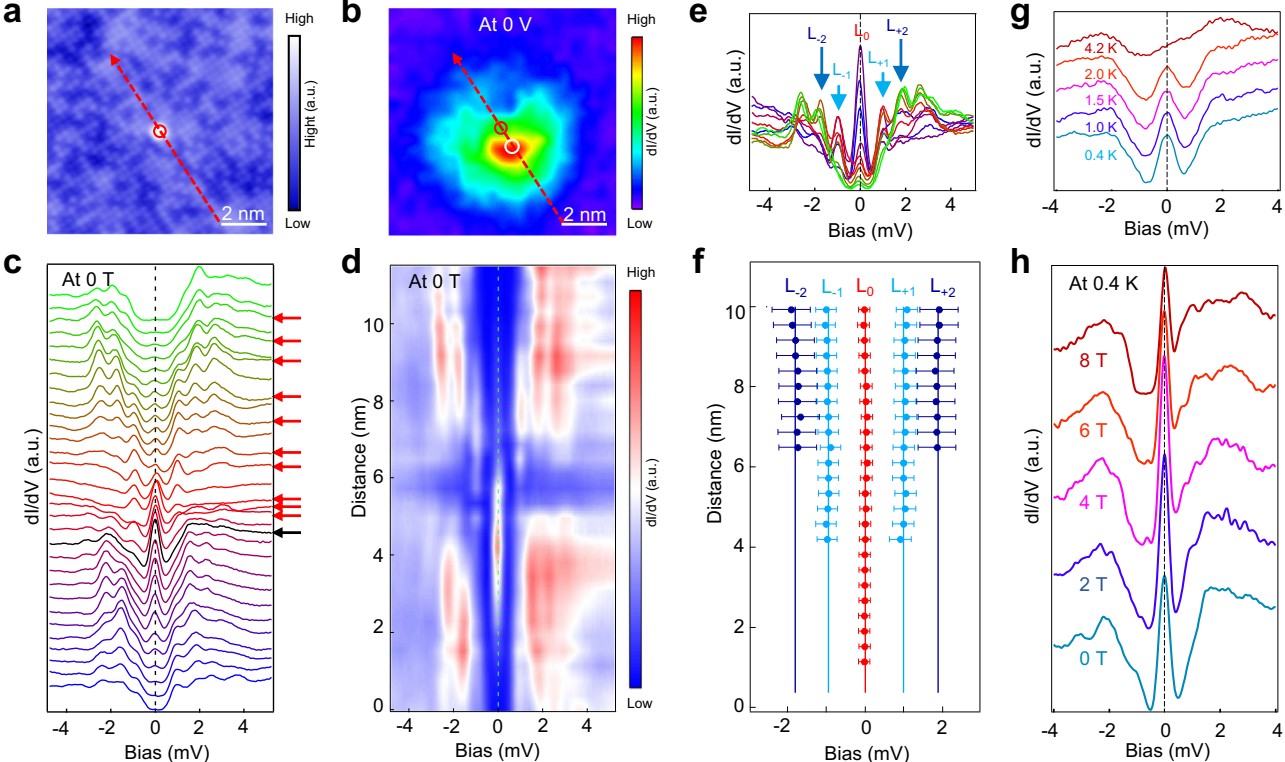

**Fig. 2 ZBP and integer quantized in-gap states on type-I Fe adatoms. a** An atomic-resolution topographic image ($V_s = -10$ mV, $I_t = 100$ pA) of a single type-I Fe adatom (red circle). **b** A zero-energy d$I$/d$V$ map ($V_s = -10$ mV, $I_t = 100$ pA) of the same area in **a**. **c** and **d** d$I$/d$V$ spectra and the corresponding intensity plot ($V_s = -10$ mV, $I_t = 200$ pA) along the line-cut indicated by the red dashed and arrowed line in **a** and **b**. The decay length of the ZBP is about 2.5 nm. **e** A stacking plot of several d$I$/d$V$ spectra ($V_s = -10$ mV, $I_t = 200$ pA) marked by the red arrows in **c**. The energies of the in-gap states are labeled as $L_0$, $L_{\pm1}$, $L_{\pm2}$. **f** The energies of the in-gap states at different spatial positions along the line-cut. Error bars are the FWHM of a Gaussian fit. The energy values of the solid lines are calculated as the average energy of the in-gap states, showing the same integer quantization as the topological vortex core states anchored by the MZM. **g** Temperature dependence of the d$I$/d$V$ spectra ($V_s = -10$ mV, $I_t = 200$ pA) measured at zero-energy intensity center (white circle in **b**) with the corresponding spectrum (the black curve) pointed by the black arrow in **c**. The ZBP intensity decreases with increasing temperature and becomes indiscernible at 4.2 K. **h** Magnetic field dependence of the d$I$/d$V$ spectra ($V_s = -10$ mV, $I_t = 200$ pA) also acquired at the zero-energy intensity center (white circle in **b**) at 0.4 K. The ZBP remains robustly at zero energy without splitting.

free YSR states no longer cross zero energy, due to the Zeeman splitting that removes the accidental degeneracy. However, the emergence of the unsplit ZBP at higher $G_N$ is unabridged even at such a high field, indicating that ZBP corresponds to a single-MZM robust against an applied magnetic field. This identification is further corroborated by performing the measurements on type-I Fe adatoms that show the ZBP associated with the MZM upholds its integrity and does not shift or split with increasing $G_N$ in a field as high as 6 T (Supplementary Note 6 and Supplementary Fig. 8). The compelling evidence attributes the novel coalescence of in-gaps states toward the ZBP to the change in the nature of the magnetic impurity-induced defect state from the vortex-free YSR state to a vortex state with a vortex MZM (Fig. 3b), which is fully consistent with the theoretical prediction that increasing the exchange coupling of an Fe impurity induces a transition from the YSR states to the QAV states hosting a MZM in FeTe$_{0.55}$Se$_{0.45}$ superconductors[28]. We note that the entire tip-manipulation process is carried out locally without affecting the stability of the superconducting topological surface states. The SC gap in the STM spectrum (Fig. 3) does not close and the transition to the QAV state is reversible. The vortex MZM naturally arises in the QAV core due to the superconducting topological surface states on the surface of FeTeSe[28]. Manipulating the Fe adatom by the approaching-tip to the $C_4$ symmetric site and closer to the superconducting surface allows the magnetic Fe to sustain its orbital magnetic moment in addition to the spin

moment and increases the spin–orbit exchange coupling, which generates the circulating supercurrents around the defect, leading to the flux trapping and the nucleation of a vortex state. The transition between the YSR states and the MZM is even reversible, as the d$I$/d$V$ spectra as a function of the barrier conductance retrace that shown in Fig. 3e, f upon withdrawing the tip (Fig. 3g, h) in a controlled manner. The transition is also replicable when the type-II Fe adatom under the tip in Fig. 3 is moved to a different location about 1 nm away (Supplementary Fig. 9). These observations reveal the unprecedented nature of defect excitations in the superconducting TSS where local magnetic moment and screening currents are inextricably connected through the magnetoelectric effect.

**Hybridization between MZMs in QAV and field-induced vortex.**
The phase coherence of the MZMs is stored nonlocally and protected by the topological degeneracy against environmental decoherence caused by local perturbations, which is the central to the idea of topological quantum computing[32,33]. The coupling of two MZMs sufficiently close by annihilates of the nonabelian anyonic zero modes and creates a pair of fermionic states at nonzero energies. This coupling process usually requires two overlapping magnetic field-induced vortices, which is difficult to control on the Abrikosov lattice[7,8,34]. Our system allows a new possibility, i.e., the hybridization between MZMs hosted in a QAV and a field-induced vortex (Fig. 4). The zero-energy d$I$/d$V$ map (Fig. 4a) shows a MZM

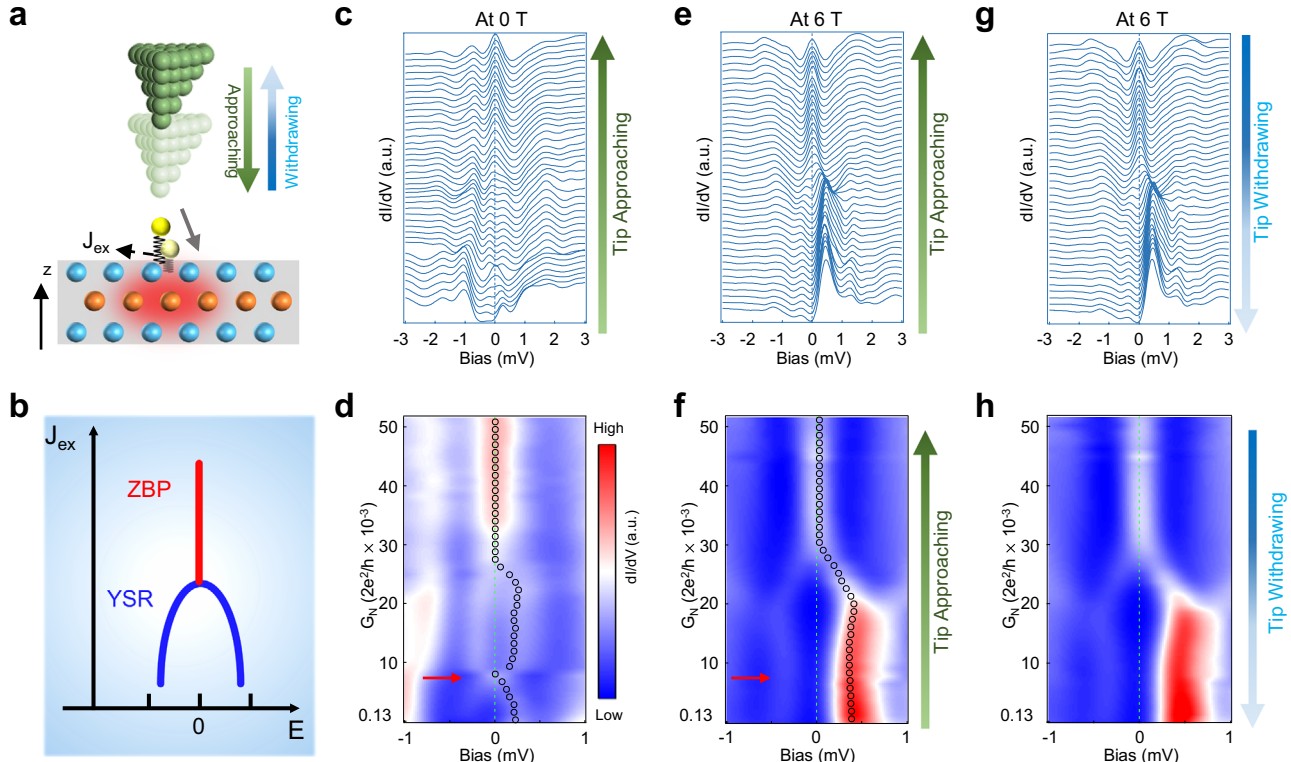

**Fig. 3 Reversible transitions between YSR states and a robust ZBP/MZM induced by modulating the exchange coupling of type-II Fe adatoms using the STM tip. a** Schematics illustrating an approaching STM tip on top of a type-II Fe adatom can move the Fe adatom in directions parallel and perpendicular to the Te/Se surface and modulate the spin–orbit exchange coupling ($J_{ex}$) to the superconductor. **b** Schematic phase diagram from the vortex-free YSR states to an anomalous vortex MZM represented by the ZBP with the increasing exchange coupling $J_{ex}$. **c** and **d** Tunnel-barrier conductance dependence of the d$I$/d$V$ spectra in **c** and its intensity plots in **d**, showing the evolution of vortex-free YSR states into a robust ZBP at 0 T under an approaching-tip. Red arrow in **d** indicates the position of an accidental near degeneracy of the YSR states. Black circles in **d** trace the peak positions. **e** and **f** are the same as **c** and **d**, but measured in a magnetic field of 6 T. Black circles in **f** trace the peak positions. The accidental degeneracy of the YSR states in 0 T in **d** is removed by the 6 T magnetic field, while the evolution to the ZBP remains robust. **g** and **h** are the same as **e** and **f**, but showing the transition from ZBP to YSR states by withdrawing the STM tip at 6 T, indicating the transition is reversible.

in the QAV nucleated at a type-I Fe adatom in a magnetic field of −0.2 T, also visible in the intensity plot (Fig. 4b) as sharp ZBPs along the line-cut across the adatom. A field-induced vortex is observed to enter the field of view subsequently. The latter sits very close to the Fe site, thus enlarges significantly the region with spectral weight at zero-energy (Fig. 4c). Remarkably, acquiring the intensity plot along the same line-cut shows that the ZBP splits into two peaks separated by an energy spacing ~0.25 meV (Fig. 4d). During the second round of the measurements, the vortex creeps away. The zero-energy map recovers (Fig. 4e) and the ZBP reemerges in the intensity plot (Fig. 4f). Throughout the hybridization process, the temperature and the magnetic field are kept stable and the change in the position of the field-induced vortex is due to the spontaneous vortex creeping. It is necessary to point out that the creeping of the vortices is observed quite often when we detected MZMs in the filed-induced vortices in our previous work. Three representative d$I$/d$V$ spectra corresponding to the three conditions are extracted and displayed in Fig. 4g for a better comparison. Repeating the measurements on the same Fe adatom in a higher magnetic field of −3 T reveals again the splitting of the ZBP caused by the presence of a nearby field-induced vortex (Fig. 4h), with a larger energy spacing ~0.35 meV (Fig. 4i), possible due to the shorter distance and stronger overlap of the two MZMs as indicated by the smaller ring feature in the zero-energy map in Fig. 4h compared to Fig. 4c. Moreover, a field-induced vortex can also enter the field of view without causing detectable splitting of the ZBP bound to the Fe adatom or the

vortex MZM when they are relatively far apart (Supplementary Note 8 and Supplementary Fig. 10). These observations further support the identification of the ZBP induced by type-I Fe atoms as the MZM and concurrently provide the first experimental evidence for the hybridization between two vortex MZMs as illustrated in Fig. 4j.

## Discussion
Our STM/S measurements on magnetic Fe adatoms deposited on the surface of FeTe$_{0.55}$Se$_{0.45}$ superconductors revealed the spontaneous formation of anomalous vortex matter with integer quantized core states and MZMs in zero external field, and the reversible transition between a YSR impurity and a QAV with increasing exchange interaction strength. These fundamental properties, previously unobserved in superconductors, are consistent with the predications of the theoretical proposal of the QAV[28]. They have broad implications for superconductors with a nontrivial topological band structure and spin-momentum locked superconducting TSS, which have also been found in several other iron-based superconductors such as LiFeAs[35,36], (Li$_{0.84}$Fe$_{0.16}$)OHFeSe[37], and CaKFe$_4$As$_4$[38]. We also noticed a recent preprint posted on the arxiv[39], where the result shows absence of spin-polarization on a ZBP. It can be included in our results of type-II Fe adatoms, where the YSR states can locate at a quantum phase transition point. This result can be clarified by measurements under a high magnetic field. Together with the observed hybridization of the MZMs in the QAV nucleated at the Fe adatom and the nearby field-induced

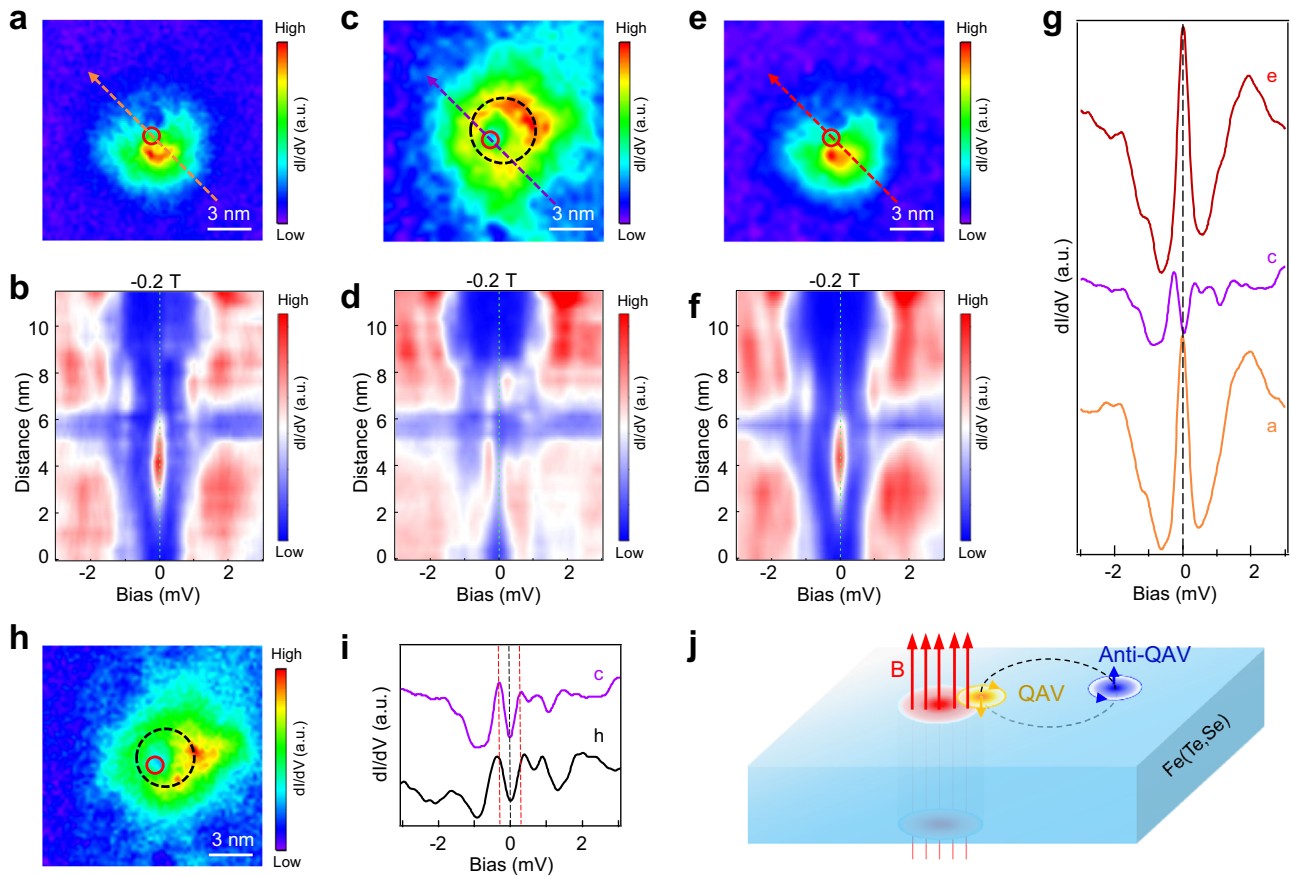

**Fig. 4 Hybridization between two MZMs in the QAV and field-induced vortex. a** Zero-energy d$I$/d$V$ map ($V_s = -10$ mV, $I_t = 100$ pA) around a type-I Fe adatom in a magnetic field of $-0.2$ T, showing the spatial distribution of the MZM in an isolated QAV nucleated at the adatom. **b** Intensity plot of d$I$/d$V$ spectra ($V_s = -10$ mV, $I_t = 200$ pA) along the line-cut indicated by the arrowed red dashed line in **a**, showing a robust ZBP. **c** and **d** are the same as **a** and **b**, but with an additional field-induced vortex pinned near the same Fe adatom visible in the zero-energy map in **c**. The ZBP disappears in the intensity plot of the d$I$/d$V$ spectra along the same line-cut in **d** and two in-gap states with an energy level spacing of 0.25 meV emerge. **e** and **f** are the same as **c** and **d**, but after the field-induced vortex creeps away. The MZM returns and the zero-energy d$I$/d$V$ map and the intensity plot of the d$I$/d$V$ spectra along the line-cut recover those in **a** and **b**. **g** Three d$I$/d$V$ spectra obtained on the Fe adatom in **a**, **c**, and **e** ($V_s = -10$ mV, $I_t = 200$ pA), respectively. **h** Zero-energy d$I$/d$V$ map around the same Fe adatom at $-3$ T ($V_s = -10$ mV, $I_t = 100$ pA), showing an additional field-induced vortex pinned close to the adatom. **i** d$I$/d$V$ spectra obtained on the Fe adatom in **h** and **c** ($V_s = -10$ mV, $I_t = 200$ pA). Both spectra show disappearance of ZBP and emergence of two in-gap states separated by an energy spacing that is larger (0.35 meV) in the case of **h** compared to **c** (0.25 meV). **j** Schematics of the interaction between two MZMs in the QAV and the field-induced vortex.

Abrikosov vortex, our findings demonstrate that magnetic adatoms coupled to the superconducting TSS provide a realistic materials platform for creating and studying the interactions between the nonabelian topological vortex MZMs.

## Methods

**Materials and measurement.** High-quality single crystals of FeTe$_{0.55}$Se$_{0.45}$ with $T_c$ of 14.5 K are grown using the self-flux method[6]. The samples are cleaved in situ and immediately transferred into a STM head. The Fe adatoms are deposited from a high-purity (99.95%) single-crystal Fe rod acquired from *ESPI METALS* to the surface of FeTe$_{0.55}$Se$_{0.45}$ at a temperature below 20 K. Before Fe deposition, we scan the surface of FeTe$_{0.55}$Se$_{0.45}$ to ensure that there is no visible interstitial Fe impurity (IFI). The STM experiments are performed in an ultrahigh vacuum ($1 \times 10^{-11}$ mbar) LT-STM systems (USM-1300s-$^3$He), which can apply a perpendicular magnetic field up to 11 T. STM images are acquired in the constant-current mode with a tungsten tip. The energy resolution calibrated on a clean Pb (111) surface is about 0.27 meV. The voltage offset calibration is followed by a standard method of overlapping points of $I$–$V$ curves. Differential conductance (d$I$/d$V$) spectra are acquired by a standard lock-in amplifier at a frequency of 973.1 Hz, under modulation voltage $V_{\text{mod}} = 0.1$ mV. Low temperature of 0.4 K is achieved by a single-shot $^3$He cryostat.

## Data availability

All relevant data are available from the corresponding authors upon request.

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

## Acknowledgements

We thank Xi Dai, Shengshan Qin, Shiyu Peng, Sankar Das Sarma for useful discussions. The work at IOP is supported by grants from the National Natural Science Foundation of China (11888101, 61888102, 11674371 and 52022105), the National Key Research and Development Projects of China (2016YFA0202300, 2018YFA0305800 and 2019YFA0308500), and the Chinese Academy of Sciences (XDB28000000, XDB07000000, 112111KYSB20160061). The work at BNL and BC is supported by grants from US DOE, Basic Energy Sciences (DE-SC0012704, DE-FG02-99ER45747).

## Author contributions

H.-J.G. and H.D. designed STM experiments. P.F., G.Q., and H.C. performed STM experiments with assistance of F.Y., Z.H., Y.X., and C.S.. J.S., R.Z., and G.G. provided samples. K.J. and Z.Q.W. provided theoretical explanations. P.F., F.Y., G.Q., and H.C. processed experimental data with input from Y.Z., G.L., L.K., W.L., and S.D.. All the authors participated in analyzing experimental data, plotting figures, and writing the manuscript. Z.Q.W., H.D., and H.-J. G. supervised the project.

## Competing interests

The authors declare no competing interests.
