## [Peer Review File · Nature Communications]

REVIEWER COMMENTS

Reviewer #1 (Remarks to the Author):

The authors report on the investigation of impurity induced sub-gap states and vortex core bound states on the surface of FeTeSe crystals using scanning tunneling microscopy and spectroscopy experiments. We note that previous experiments from this group on the surface of FeTeSe crystals revealed localized vortex core zero bias peaks under the application of an external magnetic field that were interpreted as signatures of Majorana zero modes (MZM). In this work, the authors investigate the characteristics of sub-gap states induced by Fe impurities at different adsorption sites on the sample surface, and they report the observation of localized zero bias peaks (ZBP) near Fe adatoms adsorbed at the high-symmetry sites. Analyzing the dependence of the ZBP on different experimental parameters, such as the magnetic field, the authors interpret the ZBP as the charge signature of a MZM localized to a quantum anomalous vortex (QAV), which is induced by the coupling of the Fe adatom to the superconducting topological surface state of FeTeSe. The authors report their observations to be consistent with theoretical expectations for a QAV state.

The experimental work has been competently executed and is technically correct. The experimental data are of high quality and their presentation is clear and comprehensible. The author's interpretation is consistent with their experimental data. The manuscript is clearly written and provides sufficient methodological details. The manuscript further contains a large amount of high quality data all of which are relevant to the interpretation of the experimental results. The authors have taken large statistics (more than adatoms 100 atoms studied) to demonstrate the robustness of their observations.

I can, therefore, recommend publication of this manuscript in Nature Communications after minor revisions outlined below.

Without doubt, the observation of a reversible ZBP splitting induced by a transient vortex core presented in Fig.4 is an exciting result and deserves attention. The authors should, however, remove claims about MZM fusion, because the terminology is misleading and, does not match the experimental observations. As such, the current phrasing may weaken the manuscript.

Specifically, the authors experimentally observe a spectral splitting, which is caused by MZM hybridization owing to a spatial overlap of their wavefunctions, as the field-induced vortex passes by. Similar results on the effect of MZM hybridization have been reported before on the nanowire platform (Nature volume 531, pages206–209(2016)). MZM fusion, instead, is typically associated with the deliberate annihilation of MZM pairs and the creation of fermionic modes, after braiding operations have been performed. The result of such an MZM fusion, i.e. the change of fermion number, can be measured experimentally and allows to track the braiding operations. Different schemes, involving current and charge sensing, have been proposed for that purpose (e.g. see Phys. Rev. X 6, 031016 (2016)). In sum, claims on MZM fusion should, therefore, rest on the experimental observation of changes in fermion numbers using suitable methodologies, and the observation of a spectral splitting does not warrant that claim in my understanding.

Reviewer #2 (Remarks to the Author):

This paper reports an STM study of Fe atoms on the topological superconductor FeTeSe. Earlier studies have observed Majorana zero modes in field-induced vortex cores and in quantum anomalous vortex (QAV) cores. The latter had been induced by interstitial Fe atoms/defects. Here, the authors deposit a

dilute amount of Fe atoms on the substrate and study their spectral properties by STM. They find two types of Fe atoms: type I atoms all sit in a C4 symmetry site and exhibit a zero-energy state, which is robust in an external magnetic field and interpreted in terms of a QAV. In contrast, type II atoms exhibit trivial YSR states, which shift in a B field. In $\sim 27\%$ of these cases, the authors have manipulated the states such that the YSR states shift to zero energy. The main claim is that the tip approach leads to an increase in the exchange coupling of the Fe atom with the substrate, eventually driving the system into an QAV core with a MZM. This concept is not entirely clear. A topological state should be robust and cannot be transformed smoothly to a trivial state and vice versa by simply changing the exchange coupling strength. Indeed, a topological state is characterized by its robustness against external perturbation, such as a magnetic field or a change in the exchange coupling.

Instead, there must be an abrupt different change in symmetry and interaction with the substrate. For example, the Fe atoms can move to another adsorption site/symmetry during tip manipulation – as the authors correctly point out. If the atom then falls into the adsorption site of type I atoms (which exhibit C4 symmetry and QAV state), it would exhibit the QAV state. Pushing the Fe into a different site is a well-known manipulation tool. Such a process can be considered trivial and does not merit publication in Nature Communications or should be characterized in terms of the precise changes in more detail. If they can exclude this process, they should provide an alternative mechanism.

Some additional remarks:

A more detailed characterization of the type II adsorption sites is missing.

In type I atoms, the MZM is not at the Fe center but slightly off-centered. Why? Why is the vortex not circular? What is the origin of the symmetry breaking?

Is the manipulation with the STM tip centered on the type-II Fe atom? Why does the MZM appear in this case on this very position?

In the second part of the manuscript the authors describe an interaction of a field-induced vortex core with a QAV. Here, control of the vortex motion seems to be missing or not sufficiently described.

Which parameters do the authors use to control the process?

In summary, this manuscript does not provide a significant advancement of the understanding of topological states and their manipulation. I cannot recommend publication in Nature Communications.

Point-by-point response to the comments from the reviewers

We thank the reviewers for their time and effort for reviewing our manuscript. We have addressed all the criticisms and comments point-by-point below and revised the manuscript accordingly. In this response letter, comments from the reviewers are summarized in black italic typeface. Our responses are in regular blue typeface. Our changes to the text are in red.

Response to Reviewer 1[#]

“The experimental work has been competently executed and is technically correct. The experimental data are of high quality and their presentation is clear and comprehensible. The author’s interpretation is consistent with their experimental data. The manuscript is clearly written and provides sufficient methodological details. The manuscript further contains a large amount of high quality data all of which are relevant to the interpretation of the experimental results. The authors have taken large statistics (more than adatoms 100 atoms studied) to demonstrate the robustness of their observations.

I can, therefore, recommend publication of this manuscript in Nature Communications after minor revisions outlined below.”

Response: We thank the reviewer for the positive remarks on the significance and quality of our work. In the following, according to reviewer’s suggestions, we have done further revisions to address the concerns from the reviewer, and made the revisions correspondingly.

Comment 1. *The authors should, however, remove claims about MZM fusion, because the terminology is misleading and, does not match the experimental observations. As such, the current phrasing may weaken the manuscript.*

Response 1: We have taken the reviewer’s suggestions and have replaced “fusion” by “hybridization” as follows.

In the title:

“Observation of magnetic adatom-induced Majorana vortex and its **hybridization** with field-induced Majorana vortex in an iron-based superconductor”

In the abstract:

“the realization of its **hybridization** with a nearby field-induced Majorana vortex in iron-based superconductor $\text{FeTe}_{0.55}\text{Se}_{0.45}$.”

“The magnetic field-induced Abrikosov vortex lattice makes it difficult to braid a set of Majorana zero modes or to study the **coupling** of a Majorana doublet due to overlapping wave functions.”

In the last paragraph of Results part:

“Our system allows a new possibility, i.e., the **hybridization** between MZMs hosted in a QAV and a field-induced vortex (Fig. 4).”

“Throughout the **hybridization** process, the temperature and the magnetic field are kept stable.”

“These observations further support the identification of the ZBP induced by type-I Fe atoms as the MZM and concurrently provide the first experimental evidence for the **hybridization** between two vortex MZMs as illustrated in Fig. 4j.”

In the Discussion part:

“Together with the observed **hybridization** of the MZMs in the QAV nucleated at the Fe adatom and the nearby field-induced Abrikosov vortex, our findings demonstrate that magnetic adatoms coupled to the superconducting TSS provide a realistic materials platform for creating and studying the interactions between the nonabelian topological vortex MZMs.”

In the caption of Fig. 4

“Fig. 4 **Hybridization** between two MZMs in the QAV and field-induced vortex.”

In part VIII of Supplementary Information

“These findings support the observation of **hybridization** between the two MZMs inside the Fe adatom induced topological QAV and the field-induced topological Abrikosov vortex presented in Fig. 4 of the main text.”

Response to Reviewer 2[#]

This paper reports an STM study of Fe atoms on the topological superconductor FeTeSe. Earlier studies have observed Majorana zero modes in field-induced vortex cores and in quantum anomalous vortex (QAV) cores. The latter had been induced by interstitial Fe atoms/defects. Here, the authors deposit a dilute amount of Fe atoms on the substrate and study their spectral properties by STM. They find two types of Fe atoms: type I atoms all sit in a C4 symmetry site and exhibit a zero-energy state, which is robust in an external magnetic field and interpreted in terms of a QAV. In contrast, type II atoms exhibit trivial YSR states, which shift in a B field. In ~27% of these cases, the authors have manipulated the states such that the YSR states shift to zero energy.

Comment 1. *The main claim is that the tip approach leads to an increase in the exchange coupling of the Fe atom with the substrate, eventually driving the system into an QAV core with a MZM. This concept is not entirely clear. A topological state should be robust and cannot be transformed smoothly to a trivial state and vice versa by simply changing the exchange coupling strength. Indeed, a topological state is characterized by its robustness against external perturbation, such as a magnetic field or a change in the exchange coupling. Instead, there must be an abrupt different change in symmetry and interaction with the substrate. For example, the Fe atoms can move to another adsorption site/symmetry during tip manipulation – as the authors correctly point out. If the atom then falls into the adsorption site of type I atoms (which exhibit C4 symmetry and QAV state), it would exhibit the QAV state. Pushing the Fe into a different site is a well-known manipulation tool. Such a process can be considered trivial and does not merit publication in Nature Communications or should be characterized in terms of the precise changes in more detail. If they can exclude this process, they should provide an alternative mechanism.*

Response 1:

We thank the referee for raising this important point. We overlooked the importance of a clear discussion on this point in the previous version, which may have caused the confusion. The topological property of FeTeSe lies in its nontrivial topological band structure that supports the topological surface states (TSS) above T_c , which become the superconducting TSS below T_c^1 . In our work, we study the nature of the defect excitations associated with magnetic Fe adatoms deposited on the surface of FeTeSe, all without changing this topological property of FeTeSe associated with the superconducting TSS. This is possible because these are localized excitations of the superconductor. We found two different kinds of defect excitations. One kind is the conventional vortex-free Yu-Shiba-Rusinov (YSR) bound states at the type-II Fe

adatoms. These are what expected at the magnetic impurity in a superconductor. Importantly, we observed, for the first time, a new kind of magnetic impurity induced defect states at the type-I Fe adatoms, i.e. a series of integer quantized localized bound states including one at zero energy. These defect excitations are remarkably similar to the vortex cores states in magnetic field induced Abrikosov vortices, where the bound state at zero-energy has been widely identified with the vortex MZM in the presence of the superconducting TSS^{2,3}. This led us to the highly significant finding of the theoretically proposed quantum anomalous vortex (QAV) and the MZM induced by the magnetic Fe adatoms in FeTeSe. Note that the essential point is the nucleation of a vortex state at the type-II Fe adatoms; the MZM arises naturally from the superconducting TSS. We stress that although a single in-gap state at zero-energy has been observed at the interstitial Fe impurities introduced during the growth of bulk FeTeSe⁴, our observation of the series of integer quantized bound states is crucial for the identification of the vortex nature of the QAV.

Then, we demonstrated for the first time in any superconductor that the vortex-free YSR state can be turned into the quantum anomalous vortex state with the MZM by pushing the STM tip toward a type-II Fe adatom. The entire tip-manipulation process is carried out locally without affecting the superconducting state and the topological surface state. The STM spectrum in Fig. 3 shows that the superconducting gap does not close as the YSR states at nonzero energies are driven to zero energy by the approaching tip, and the transition to the QAV state is reversible. The superconducting TSS is thus stable against the perturbation and there is no topological phase transition during the tip-manipulation process. The effect of the tip manipulation is to increase the local exchange coupling between the magnetic Fe impurity and the superconducting quasiparticles and drive a transition in the nature of the magnetic defect excitation from the vortex-free YSR state to a vortex state. This transition was predicted in the theory the QAV⁵, where it was shown that the vortex MZM naturally arises in the QAV core due to the already present superconducting TSS in FeTeSe.

The key requirement for generating a QAV in a superconductor with strong spin-orbit coupling is a strong enough exchange coupling between the magnetic moment and the superconducting quasiparticles⁵. The high symmetry sites, such as the C4 symmetric sites, allows the magnetic Fe to develop its orbital magnetic moment in addition to the spin moment⁵. Moving the Fe adatom by the approaching tip to the C4 symmetric site and closer to the superconducting surface thus increases the magnetic moment and the spin-orbit exchange coupling, which generates the circulating supercurrents around the defect, leading to flux trapping and the nucleation of a vortex state. Hence, the tip approaching induces a defect state transition from the YSR state to the QAV state.

In view of the referee's comment, we have added discussions in the seventh paragraph in the Results part:

“The compelling evidence attributes the novel coalescence of in-gaps states toward the ZBP to the change in the nature of the magnetic impurity-induced defect state from the vortex-free YSR state to a vortex state with a vortex MZM (Fig. 3b), which is fully consistent with the theoretical prediction that increasing the exchange coupling of an Fe impurity induces a transition from the YSR states to the QAV states hosting a MZM in $\text{FeTe}_{0.55}\text{Se}_{0.45}$ superconductors²⁸. We note that the entire tip-manipulation process is carried out locally without affecting the stability of the superconducting topological surface states. The SC gap in the STM spectrum (Fig. 3) does not close and the transition to the QAV state is reversible. The vortex MZM naturally arises in the QAV core due to the superconducting topological surface states on the surface of FeTeSe ²⁸. Manipulating the Fe adatom by the approaching tip to the C_4 symmetric site and closer to the superconducting surface allows the magnetic Fe to sustain its orbital magnetic moment in addition to the spin moment and increases the spin-orbit exchange coupling, which generates the circulating supercurrents around the defect, leading to the flux trapping and the nucleation of a vortex state”.

Comment 2. A more detailed characterization of the type II adsorption sites is missing.

Response 2: We have characterized the adsorption site of the type II Fe adatom in the paper. In view of the referee's comment, which we agree, we have revised Fig. S7a accordingly and provided additional information that the adsorption sites of the type II Fe atoms can be on or off the C_4 symmetric site as follows:

Revised Supplementary Fig. S7. Modulating YSR states with approaching STM tip at a type-II Fe adatom. **a**, A high-resolution STM image showing a single type II Fe adatom located away from the C_4 symmetric site. (The C_4 symmetric site is highlighted by the vertices of the white dashed grid). **b**, The tip-sample distance offset z versus tunnel-barrier conductance G_N as the tip approaches the adatom. The z -offset decreases smoothly with increasing barrier conductance G_N . **c** and **d**, Normalized dI/dV spectra and the corresponding intensity plot under different tunnel-barrier conductance in zero applied magnetic field, showing the shift of YSR states. The red arrow indicates the quantum transition point of the YSR states.

Comment 3. In type I atoms, the MZM is not at the Fe center but slightly off-centered. Why? Why is the vortex not circular? What is the origin of the symmetry breaking?

Response 3: We thank the referee for this comment. This is indeed another important property of the magnetic adatom induced QAV and vortex MZM discovered by our experiments that was not previously known, and we should have provided a discussion of its origin. The symmetry breaking originates from the orientation of the magnetic moment of the Fe adatom. When a Fe adatom is adsorbed on the surface of $\text{FeTe}_{0.55}\text{Se}_{0.45}$, its magnetic moment is not necessarily pointing normal to the plane. When the magnetic moment is normal to the plane, the vortex and the MZM are centered at the magnetic impurity because of the rotation symmetry as shown in the accompanying Fig. R1a below. However, when the magnetic moment is canted away from the surface normal such that it contains an in-plane component, the rotation

symmetry is broken because of spin-orbit coupling. Both the vortex and the MZM are off-centered as shown in Fig. R1b. More detailed explanation of these remarkable properties and explicit calculations will be given in a separate theoretical paper which is being written up. We included here these results for the referee.

Fig. R1 Zero-energy density of states maps for QAVs with different orientations of the magnetic moment obtained by BdG calculations on a disc. **a.** The magnetic moment is along the z direction. **b.** The magnetic moment points 45° degrees away from the z axis.

We have added the following explanation in the third paragraph of Results part and provided a citation to our theory work. If the referee feels that it is necessary to include the above figure, we will be happy to do so.

“... with the zero-energy intensity center slightly offset from the Fe site. The breaking of the rotation symmetry is likely due to the canting of the magnetic moment of the Fe adatom away from the surface normal and the presence of spin-orbit coupling²⁹.”

Comment 4. *Is the manipulation with the STM tip centered on the type-II Fe atom? Why does the MZM appear in this case on this very position?*

Response 4: Yes, the STM tip is centered on top of the type-II Fe atom and detects a spectral peak at zero-energy. We need to emphasize that, in the manipulation experiments with the STM tip, we do not know whether this zero-mode is centered at the Fe adatom or not, since in this case it is impossible to move the STM tip to obtain a spatial map while keeping the position of the Fe adatom being manipulated frozen. Whether the MZM is centered on or off the manipulated Fe adatom will not affect the observation of the spectral peak at zero-energy, as can be seen from the case of the type-I Fe adatom shown in Fig. 2b.

To clarify this point, we have added a discussion in the sixth paragraph of the Results part:

“These observations motivate us to manipulate the exchange coupling between the magnetic adatoms and the substrate by tuning the tip to sample distance^{29, 30}. In the approaching-tip process, the STM tip needs to be positioned on top of the Fe adatom, which makes it impossible to acquire spatial maps while keeping the position of the Fe adatom frozen.”

Comment 5. In the second part of the manuscript the authors describe an interaction of a field-induced vortex core with a QAV. Here, control of the vortex motion seems to be missing or not sufficiently described. Which parameters do the authors use to control the process?

Response 5: We thank the referee for raising this question. In our STM experiments, we did not control or manipulate the vortex motion. The movement of the vortex results from the spontaneous vortex creeping. Fig. R2 shows a typical example of spontaneous vortex creeping that we observed occasionally in the experiments. We acquired three zero-bias conductance maps in the same region but at different times, as shown in Fig. R2(b-d), respectively. Although we did not change any parameter during the measurements, the spontaneous change in the positions of the three vortices was clearly observed.

Fig. R2 Spontaneously vortex creeping at 4 T. **a**, Topographic STM image of the surface region of FeTe_{0.55}Se_{0.45}. **b-d** Zero-energy maps of the field-induced vortices at three different times in the same area of **a**.

In view of the referee's comment, we have added a clarifying sentence in the last paragraph of the Results part.

“Throughout the hybridization process, the temperature and the magnetic field are kept stable and the change in the position of the field-induced vortex is due to the spontaneous vortex creeping”.

Overall Critical Comment:

In summary, this manuscript does not provide a significant advancement of the understanding of topological states and their manipulation. I cannot recommend publication in Nature Communications.

Response: While we appreciate the comments and suggestions of Referee 2, we cannot agree with the referee claim that our manuscript does not provide a significant advancement. We want to clarify once again the novelty and potential impact of our findings both for advancing the basic physical understanding of defect excitations in superconductors, and for the manipulation of defect states such as YSR, quantum anomalous vortex, and vortex MZMs. Our work contains three original contributions.

- (a) We reported for the first observation in any superconductor the integer quantized in-gap states localized at magnetic adatoms. It provides novel and indispensable evidence for the existence of QAV through observations of the vortex core states.
- (b) We reported for the first time the transition in the nature of the defect excitations in a superconductor from the YSR state to the QAV state, when the spin-orbital exchange coupling of the magnetic adatom is manipulated by the STM tip. This phenomenon provides novel insights into the unexpected properties of localized quantum states in superconductors with topological surface states.
- (c) We demonstrated for the first time that the dual origins of the vortex MZM in the QAVs nucleated at magnetic adatoms and the in the field-induced vortices allow the unprecedented study of the hybridization and annihilation of two MZMs. This offers a possible new direction for studying the interactions and fusion of the MZMs in the future, in connection to topological quantum information processing.

We thank the reviewer for the comments and suggestions. We have fully addressed the reviewer's concerns by adding theoretically analysis and discussions, and correspondingly revised the manuscript and Extended Data Figures. We believe that all the changes made in the response and the revisions have strengthened our work greatly.

1. Zhang P *et al.* Observation of topological superconductivity on the surface of an iron-based superconductor. *Science* **360**, 182-186 (2018).
2. Liu Q *et al.* Robust and clean Majorana zero mode in the vortex core of high-temperature superconductor $(\text{Li}_{0.84}\text{Fe}_{0.16})\text{OHFeSe}$. *Phys Rev X* **8**, 041056 (2018).
3. Kong L *et al.* Half-integer level shift of vortex bound states in an iron-based superconductor. *Nat Phys* **15**, 1181-1187 (2019).
4. Yin JX *et al.* Observation of a robust zero-energy bound state in iron-based superconductor Fe(Te,Se). *Nat Phys* **11**, 543-546 (2015).
5. Jiang K, Dai X, Wang ZQ. Quantum anomalous vortex and Majorana zero mode in iron-based superconductor Fe(Te,Se). *Phys Rev X* **9**, 011033 (2019).

REVIEWERS' COMMENTS

Reviewer #1 (Remarks to the Author):

The authors have addressed my comments, and I can, therefore, recommend publication in Nature Communications.

Nevertheless, I would like to direct their attention to a recent preprint posted on the arxiv (<https://arxiv.org/abs/2011.05470>), which reports spin-polarized studies on the Fe adatoms on the FeTeSe surface. Based on the absence of a spin-polarization for the zero-bias peak observed on Type-I adatoms, the authors of this preprint conclude that the ZBP cannot be interpreted as a signature of a MZM. Even though experimental results and their interpretations may vary, I strongly recommend the authors to consider this preprint in their discussion, because a most up-to-date literature list would only strengthen the manuscript.

Reviewer #2 (Remarks to the Author):

The authors have submitted a revised version of the manuscript and tried to answer my questions. I am still confused about the tip position and the appearance of the MZM. The localization of the MZM is apparently not directly on top of the Fe atom, but delocalized next to the Fe atom (Fig. 2b). Hence, if the tip is approached directly on the Fe atom, one should not be able to observe such a strong MZM. This appears as a contradiction to me. The only possible explanation is a larger lateral misplacement of the Fe atom upon tip approach, as I suggested in my previous report. Unfortunately, the authors did not rule out this scenario explicitly, though they admit some lateral displacement, which supposedly induces sufficient exchange coupling and spin orbit coupling. Fig. R1 provides important information on the angular distribution. However, the length scale is missing. Does it agree with the spatial extent in the STM images? Once, my first question is solved, I suggest that this spatial distribution may also be included in the manuscript.

Report on Nature Communications manuscript NCOMMS-20-35832A “Observation of magnetic adatom-induced Majorana vortex and its hybridization with field-induced Majorana vortex in an iron-based superconductor” by P. Fan et al.

This paper reports on a thorough investigation of Fe atoms adsorbed on the topological superconductor FeTeSe. The original findings include i) the observation of integer quantized in-gap states localized on the magnetic adatoms, demonstrating the presence of a quantum anomalous vortex (QAV), ii) the transition in the nature of the defect excitations from YSR state to the QAV state via tip-sample interaction, iii) and the hybridization of a QAV with a magnetic field induced vortex resulting in the transition from integer quantized to half-integer quantized (topological) in-gap states.

The paper is well written, easy to follow with a very clear presentation of the data that are accompanied with high quality supplementary materials. The whole study has been performed in a systematic way with an impressive amount of data collected (100 atoms studied), making all the experimental observations extremely robust. Each adatom types (I and II) have been systematically studied as a function of temperature, magnetic field and tip-to-sample distance, demonstrating a clear reproducibility. In the whole presentation I don't see any flaw in the data presentation. Perhaps, in the section concerning the hybridization of the QAV with the magnetic field induced vortex, the authors could show a larger image showing part of the vortex lattice to demonstrate that a magnetic field induced vortex does sit close to the QAV location. This would complement Fig. 4c and provide unambiguous evidence for their scenario. Apart from that remark, I think the paper is of very high quality and deserves publication in Nature Communications.

Comment on Rebuttal

The authors replied point-by-point to all questions raised by the Reviewers. Reviewer #1 recommended the manuscript for publication. Reviewer #2 however raised 5 comments and claimed that this work does not provide a significant advancement. The authors provide very convincing answers to comments 2 to 5, accompanied with additional data and simulations. The response to comment 1 is less clear. In this comment 1 Reviewer #2 suspects that, in the tip-induced transition from YSR to QAV, the Fe adatom is actually moving away to its initial adsorption site upon approaching the tip, due to the tip-to-sample interaction. This is indeed a reasonable scenario. However it is very difficult experimentally (if not impossible) to demonstrate that the adatom does not indeed move. Nevertheless, given that the authors show in Supp. Section III and Supp. Fig S4 that the type I site is more stable (after annealing), it seems unlikely that, assuming the adatom moves with approaching the tip, a type II adatom, which transforms into a type I upon approaching the tip, does not stay type I when retracting the tip. The fact that the process is reversible points to an absence of adatom displacement in agreement with the author interpretation.

Furthermore Reviewer 2 argues that a topological state is, by essence, robust with respect to external perturbations and that it cannot be changed by tuning continuously an external parameter, such as J_{ex} . Therefore, a transition to a topological state would necessarily imply an abrupt symmetry change, as hopping of the adatom to a different adsorption site. This argument is however not correct. The literature is full of phase diagrams demonstrating the transition between topologically trivial and non-

trivial states, which can be continuously tuned, for example by the Zeeman field V_Z or the superconducting triplet amplitude Δ_T .

Therefore comment 1 cannot justify the rejection of this manuscript.

Minor comments

- Abstract: the sentence "Here we report ..." is very long and difficult to follow. It includes several concepts, some not introduced in the two first sentences (i.e. QAV) and two main results (observations). The authors should work out this and perhaps split it in two.

- §1 p4: change "at varies heights" to "various heights"

- §2 p6: change "perform the measurement again at 0.4K" to "performing [...]"

- §2 p6 remove "in a captivating manner"; §2 p7 remove "the ravages of". Clichéd expressions...

Point-by-point response to the comments from the reviewers

We thank the reviewers for their effort for reviewing our manuscript and providing us valuable suggestions and comments. We have addressed all the questions and comments point-by-point below and revised the manuscript accordingly. In this response letter, comments from the reviewers are summarized in black italic typeface. Our responses are in regular blue typeface. Our changes to the text are in red.

Response to Reviewer 1[#]

“The authors have addressed my comments, and I can, therefore, recommend publication in Nature Communications.”

Response: We thank the reviewer for recommending publication of our manuscript on Nature communications.

Comment 1. Nevertheless, I would like to direct their attention to a recent preprint posted on the arxiv (<https://arxiv.org/abs/2011.05470>), which reports spin-polarized studies on the Fe adatoms on the FeTeSe surface. Based on the absence of a spin-polarization for the zero-bias peak observed on Type-I adatoms, the authors of this preprint conclude that the ZBP cannot be interpreted as a signature of a MZM. Even though experimental results and their interpretations may vary, I strongly recommend the authors to consider this preprint in their discussion, because a most up-to-date literature list would only strengthen the manuscript.

Response 1: We have taken the reviewer’s suggestions and discuss the results of the recent preprint in the Discussion part:

“CaKFe₄As₄³⁸. We also noticed a recent preprint posted on the arxiv³⁹, where the result shows absence of spin-polarization on a ZBP. It can be included in our results of type II Fe adatoms, where the YSR states can locate at a quantum phase transition point. This result can be clarified by measurements under a high magnetic field. Together with the observed hybridization”

Response to Reviewer 2[#]

The authors have submitted a revised version of the manuscript and tried to answer my questions.

I am still confused about the tip position and the appearance of the MZM. The localization of the MZM is apparently not directly on top of the Fe atom, but delocalized next to the Fe atom (Fig. 2b). Hence, if the tip is approached directly on the Fe atom, one should not be able to observe such a strong MZM. This appears as a contradiction to me. The only possible explanation is a larger lateral misplacement of the Fe atom upon tip approach, as I suggested in my previous report. Unfortunately, the authors did not rule out this scenario explicitly, though they admit some lateral displacement, which supposedly induces sufficient exchange coupling and spin orbit coupling.

Fig. R1 provides important information on the angular distribution. However, the length scale is missing. Does it agree with the spatial extent in the STM images? Once, my first question is solved, I suggest that this spatial distribution may also be included in the manuscript.

Response: Since reviewer 3 disagreed with reviewer 2 and supported our point of view, we do not address the comments of reviewer 2.

Response to Reviewer 3[#]

“This paper reports on a thorough investigation of Fe atoms adsorbed on the topological superconductor FeTeSe. The original findings include i) the observation of integer quantized in-gap states localized on the magnetic adatoms, demonstrating the presence of a quantum anomalous vortex (QAV), ii) the transition in the nature of the defect excitations from YSR state to the QAV state via tip-sample interaction, iii) and the hybridization of a QAV with a magnetic field induced vortex resulting in the transition from integer quantized to half-integer quantized (topological) in-gap states. The paper is well written, easy to follow with a very clear presentation of the data that are accompanied with high quality supplementary materials. The whole study has been performed in a systematic way with an impressive amount of data collected (100 atoms studied), making all the experimental observations extremely robust. Each adatom types (I and II) have been systematically studied as a function of temperature, magnetic field and tip-to-sample distance, demonstrating a clear reproducibility. In the whole presentation I don't see any flaw in the data presentation.”

We thank the reviewer for high remarks for our manuscript.

***Comment 1.** Perhaps, in the section concerning the hybridization of the QAV with the magnetic field induced vortex, the authors could show a larger image showing part of the vortex lattice to demonstrate that a magnetic field induced vortex does sit close to the QAV location. This would complement Fig. 4c and provide unambiguous evidence for their scenario. Apart from that remark, I think the paper is of very high quality and deserves publication in Nature Communications.*

Response 1: The suggestion is very good. However, since the time of hybridization is too short (about forty minutes), we did not have enough time to acquire the map of a large area.

Comment on Rebuttal

The authors replied point-by-point to all questions raised by the Reviewers. Reviewer #1 recommended the manuscript for publication. Reviewer #2 however raised 5 comments and claimed that this work does not provide a significant advancement. The authors provide very convincing answers to comments 2 to 5, accompanied with additional data and simulations. The response to comment 1 is less clear. In this comment 1 Reviewer #2 suspects that, in the tip-induced transition from YSR to QAV, the Fe adatom is actually moving away to its initial adsorption site upon approaching the tip, due to the tip-to-sample interaction. This is indeed a reasonable scenario. However it is very difficult experimentally (if not impossible) to demonstrate that the

adatom does not indeed move. Nevertheless, given that the authors show in Supp. Section III and Supp. Fig S4 that the type I site is more stable (after annealing), it seems unlikely that, assuming the adatom moves with approaching the tip, a type II adatom, which transforms into a type I upon approaching the tip, does not stay type I when retracting the tip. The fact that the process is reversible points to an absence of adatom displacement in agreement with the author interpretation.

Furthermore Reviewer 2 argues that a topological state is, by essence, robust with respect to external perturbations and that it cannot be changed by tuning continuously an external parameter, such as J_{ex} . Therefore, a transition to a topological state would necessarily imply an abrupt symmetry change, as hopping of the adatom to a different adsorption site. This argument is however not correct. The literature is full of phase diagrams demonstrating the transition between topologically trivial and nontrivial states, which can be continuously tuned, for example by the Zeeman field V_Z or the superconducting triplet amplitude Δ_T .

Therefore comment 1 cannot justify the rejection of this manuscript.

Response 2: We appreciate the reviewer's comments and agree with his response.

Minor comments

Comment 3. Abstract: the sentence “Here we report ...” is very long and difficult to follow. It includes several concepts, some not introduced in the two first sentences (i.e. QAV) and two main results (observations). The authors should work out this and perhaps split it in two.

Response 3: We have divided the sentence into two as follows:

“overlapping wave functions. Here we report the observation of the proposed quantum anomalous vortex with integer quantized vortex core states and Majorana zero mode induced by magnetic Fe adatoms deposited on the surface. We observe the realization of its hybridization with a nearby field-induced Majorana vortex in iron-based superconductor $\text{FeTe}_{0.55}\text{Se}_{0.45}$. We also observe”

Comment 4. §1 p4: change “at varies heights” to “various heights”

Response 4: We have changed “at varies height” to “various heights” as follows:

“distributed at various heights above”

Comment 5. §2 p6: change “perform the measurement again at 0.4K” to “performing [...]”

Response 5: We have changed “perform the measurement again at 0.4K” to “performing [...]” as follows:

“sample to 15 K and **performing** the measurement again at 0.4 K”

Comment 6. §2 p6 remove “in a captivating manner”; §2 p7 remove “the ravages of”.
Clichéd expressions...

Response 6: We have removed “in a captivating manner” in page 6 and “the ravages of” in page 7.